# PARP Inhibitor Olaparib Causes No Potentiation of the Bleomycin Effect in VERO Cells, Even in the Presence of Pooled ATM, DNA-PK, and LigIV Inhibitors

**DOI:** 10.3390/ijms21218288

**Published:** 2020-11-05

**Authors:** Valentina Perini, Michelle Schacke, Pablo Liddle, Salomé Vilchez-Larrea, Deborah J. Keszenman, Laura Lafon-Hughes

**Affiliations:** 1Instituto de Investigaciones Biológicas Clemente Estable (IIBCE), Departamento de Genética, Montevideo 11.600, Uruguay; valeperini95@gmail.com (V.P.); michimefuz@gmail.com (M.S.); pabloliddle@googlemail.com (P.L.); 2Instituto de Investigaciones en Ingeniería Genética y Biología Molecular “Dr. Héctor N. Torres”, Consejo Nacional de Investigaciones Científicas y Técnicas, Ciudad Autónoma de Buenos Aires 1428, Argentina; vilchez.ingebi@gmail.com; 3Laboratorio de Radiobiología Médica y Ambiental, Grupo de Biofisicoquímica, Centro Universitario Regional Litoral Norte, Universidad de la República (UdelaR), Salto 50.000, Uruguay

**Keywords:** poly(ADP-ribosylation), PARP-1, Olaparib, KU55933, KU-0060648, SCR7 pyrazine, VERO cells, synergism, resistance, CDKN2A

## Abstract

Poly(ADP-ribosyl)polymerase (PARP) synthesizes poly(ADP-ribose) (PAR), which is anchored to proteins. PAR facilitates multiprotein complexes’ assembly. Nuclear PAR affects chromatin’s structure and functions, including transcriptional regulation. In response to stress, particularly genotoxic stress, PARP activation facilitates DNA damage repair. The PARP inhibitor Olaparib (OLA) displays synthetic lethality with mutated homologous recombination proteins (BRCA-1/2), base excision repair proteins (XRCC1, Polβ), and canonical nonhomologous end joining (LigIV). However, the limits of synthetic lethality are not clear. On one hand, it is unknown whether any limiting factor of homologous recombination can be a synthetic PARP lethality partner. On the other hand, some BRCA-mutated patients are not responsive to OLA for still unknown reasons. In an effort to help delineate the boundaries of synthetic lethality, we have induced DNA damage in VERO cells with the radiomimetic chemotherapeutic agent bleomycin (BLEO). A VERO subpopulation was resistant to BLEO, BLEO + OLA, and BLEO + OLA + ATM inhibitor KU55933 + DNA-PK inhibitor KU-0060648 + LigIV inhibitor SCR7 pyrazine. Regarding the mechanism(s) behind the resistance and lack of synthetic lethality, some hypotheses have been discarded and alternative hypotheses are suggested.

## 1. Introduction

Poly(ADP-ribosylation) or PARylation is a post-translational protein modification that is catalyzed by poly(ADP-ribosyl)polymerases (PARPs). PARPs consume nicotinamide adenine dinucleotide (NAD+) and release nicotinamide (Nam) every time they add ADP-ribose monomers to originate a lineal or ramified chain of up to 400 residues that is covalently anchored to a single amino acid. Such a nucleic-acid-like polymer, called poly(ADP-ribose) or PAR, is strongly negatively charged since it harbors two phosphate groups per residue. PAR is recognized by PAR-binding protein domains (e.g., a macrodomain) and facilitates the assembly of multiprotein complexes through non-covalent interactions. For this reason, PAR has been called a “glue” [1]. PAR synthesis is stimulated by genotoxic insults, increasing up to 500 times overall or 7 to 8 times on specific proteins [2,3].

Both endogenous metabolism and exogenous genotoxic agents induce different types of DNA damage, such as oxidative lesions, single-base modifications, crosslinks, local changes in DNA structure, single-strand breaks (SSBs), and double-strand breaks (DSBs), which can be repaired by different DNA repair systems while the cell cycle is coordinately slowed down. Directly or indirectly generated DSBs are the most challenging type of DNA damage for the cell because they can lead to mutations, carcinogenesis, or cell death. DSBs are marked by the spread of the phosphorylation of the histone H2AX on Ser139 by the canonical kinases ATM, ATR, or DNA-PK [4] through a feed-forward signaling loop to give γH2AX foci. A less studied chromatin-bound kinase called VRK1 can also give rise to γH2AX foci, at least in response to DNA damage induced by γ-rays [5]. Lesions are then repaired either after DNA replication by homologous recombination (HR) or throughout the cell cycle by canonical nonhomologous end-joining (c-NHEJ) or alternative NHEJ, also called microhomology-mediated end joining (MMEJ). Unrepaired damage may lead to replication fork stalling and single-stranded DNA exposure, activating ATR [6]. Regarding genomic stability, it is accepted that HR is the DSBs repair pathway that allows for the highest fidelity, MMEJ is the one promoting the highest genomic instability, and c-NHEJ is in between. C-NHEJ repair occurs with a half-time of 10–30 min after damage, while MMEJ processing has a half-time of 30 min to 20 h, being considered a rescue pathway [7]. Some molecular actors have been identified in each pathway. C-NHEJ is dependent on KU70/80, DNA-PK, and LigIV. C-NHEJ is promoted by 53BP1 and antagonized by BRCA1. In contrast, DNA end-resection by MRE11 exonuclease activity prevents c-NHEJ and promotes HR with BRCA-1 and BRCA-2 recruitment (in the presence of an undamaged template) or allows for MMEJ. The latter does not involve DNA-PK, 53BP1, or LigIV but is dependent on LigIII instead. These pathways are quite complex and MMEJ is still poorly understood [7,8,9].

The PARP family has 18 members in humans. Some of them are enzymatically inactive; others act just as mono-(ADP-ribosyl) transferases and, to our knowledge, only four of them synthesize PAR [10], namely, the canonical ancient PARP-1, PARP-2, and two tankyrases (TNKS-1 and TNKS-2). PARP-1 is the most studied member of the family and is involved in the PARP response to stress, inflammation, or genotoxic insults [11,12].

The interpreted role for PARP-1 in the DNA damage response (DDR) has changed over time in the literature. In the 20 years between 1985 and 2005, there were enthusiastic positive results with PARP inhibitors (PARPis). In different human and rodent normal or transformed cell types, diverse PARPis (3AB, INO1001, E7016, AG14361, 4-ANI) applied before, during, or after DNA damage induction sensitized cells to different genotoxic agents, including ultraviolet-C radiation (UV-C), methylmethanosulfonate (MMS), ionizing radiation (IR), and the radiomimetic agent bleomycin (BLEO) [13,14,15,16,17,18,19,20,21,22,23,24]. The variable sensitization effects, i.e., additive or synergistic, suggested different biological interpretations and putative clinical implications [25]. Still, this data was promising at least as a way to reduce chemotherapy doses and consequently diminish side effects. Such results did also push the field to go on and try to understand the mechanisms involved.

Interestingly, nowadays, a paradigm shift has occurred. The current synthetic lethality paradigm states that PARPis do not appreciably affect normal cells but do potentiate DNA damage effects in cells that have certain DNA repair pathways that are blocked by the alteration (by mutation or enzymatic inhibition) of a crucial actor of the pathway. This paradigm has huge clinical implications, opening the possibility that PARPis could be used to selectively kill aberrant cells while keeping normal cells. In fact, it was under this paradigm that Olaparib (OLA, Lynparza, AZD-2281) reached U.S. Food and Drug Administration (FDA) approval to treat BRCA-1 mutated cancers (ucm572143, ucm592357, and NCT02987543). Interestingly, BRCA-1 is not only crucial for HR but also regulates BER [26,27], and OLA displays synthetic lethality with mutated BER proteins, such as XRCC1 or Polβ [28,29] and with mutated LigIV in c-NHEJ [30]. Last but not least, PARPis can affect not only DNA repair but also modulate gene transcription [31]. In fact, the potential clinical impact of PARPis is not limited to cancer therapy but also reaches many pathologies that are characterized by inflammation [11,32,33,34]. Interestingly, the major constraint of anti-cancer BLEO therapy is the early inflammation of lung parenchyma, leading to pulmonary fibrosis [35]. Thus, we reasoned that PARPis might concomitantly potentiate BLEO-directed effects in DNA-repair-hampered cells while diminishing inflammation-related side effects in normal cells.

Not only PARP-1 but also PARP-2 [36,37,38], the mono(ADP-ribosyl)transferase PARP-3 [39,40,41], and TNKS [42] are involved in the cellular response to DNA damage, particularly DSBs. OLA was initially considered a potent and specific PARP-1 inhibitor with an in vitro IC_50_ (ivitr-IC_50_) of 5 nM [43]. Later, an even lower IC_50_ was determined using full-length PARP-1 (ivitr-IC_50_: 1.4 nM), while it was demonstrated that it is also a potent PARP-2 (ivitr-IC_50_: 12 nM) [44] and PARP-3 inhibitor (ivitr-IC_50_: 4 nM) [41] but is not a potent TNKS-1 or TNKS-2 inhibitor (ivitr-IC_50_: 1230 nM and 2340 nM, respectively) [44]. As PARP-1, PARP-2, and PARP-3 are involved in DDR, the OLA effects include DNA repair hampering due to combined enzymatic inhibition/trapping of these enzymes [38,41,45,46].

VERO is a kidney epithelial cell line from a female African green monkey. It is a challenging cell culture model to use when analyzing resistance mechanisms because it preserves an epithelial morphology but can form colonies in agar and includes a subpopulation that is BLEO-resistant, according to Terasima et al. [47]. It is not tumorigenic, except at very high passages [48]. Moreover, its genome has been studied [49]. Finally, VERO cells are infection-prone [49,50,51], thus allowing studies of DNA damage, inflammation, PARylation, or epithelial-to-mesenchymal transition induction in the host cell by viral, bacterial, or parasitic infections.

In the present work, we addressed the question of whether OLA would potentiate DNA damage induced by BLEO in VERO cells. As a further step, we asked whether the simultaneous inhibition of ATM, DNA-PK, and LigIV (to block at least HR and c-NHEJ) would show synthetic lethality with the OLA effect in VERO cells. It did not. A hypothesis regarding the mechanisms involved is discussed as a guide for future research.

## 2. Results

### 2.1. OLA Did Not Potentiate BLEO Effects on VERO Cells

As a BLEO treatment has to be performed in the absence of FBS, in all experiments involving BLEO (regardless of the presence or absence of inhibitors), all the cells, including those with and without BLEO, were subjected to a 45 min serum depletion. Therefore, all differences could be attributed to BLEO itself rather than to this short serum depletion.

Previous experiments were carried out in order to choose appropriate experimental conditions (Figure A1A–D).

Cell viability dose-response curves with BLEO alone (45 min pulses) were evaluated immediately after treatment (t = 0 h), or 24 or 72 h later (Appendix A
Figure A1A). The BLEO toxicity increased slightly above 40 µg/mL. Neither a BLEO concentration increase up to 500 µg/mL nor a treatment duration of 270 min (Figure A1B, evaluation 24 h post treatment) appreciably decreased the viability. In agreement with the report by Terasima et al. from 1972 [47], the VERO cell population was composed of a BLEO-sensitive and a BLEO-resistant subpopulation.

Experiments that involved adding the BLEO hydrolase inhibitor E-64 before (30 min) and during (45 min) BLEO treatment were carried out to exclude the possibility of the existence of this BLEO-degradative enzyme in our model (Figure A1C). The lack of an effect of E-64 indicated that the resistance was not due to the expression of BLEO hydrolase.

The toxicity of OLA alone was analyzed on VERO cells (Figure A1D). An approximately 70–80% cell viability was preserved, even after a 400 nM continuous 24 or 72 h treatment. As we wanted to analyze the specific effects on PARPs involved in DDR (PARP-1/2 and 3) and we knew that low concentrations were enough to inhibit at least PARP-1 in this cell line [43], we decided to regularly use 50 nM OLA in the combined experiments. Furthermore, a 72 h evaluation point was considered to be the most informative one in our MTT assays.

The cell viability was then assayed (Figure 1A) after a 45 min serum depletion, without (control; BLEO = 0 µg/mL) or with BLEO (BLEO = 40 or 160 µg/mL), without (violet bars) or with (light blue bars) a 50 nM OLA treatment. OLA exposure involved continuous co- and post-treatment that finished at the moment of evaluation with the MTT assay, 72 h post the BLEO treatment.

As shown in Figure 1A, there was no significant difference in cell viability that was attributable to 50 nM OLA (light blue bars) in basal (BLEO = 0) or BLEO-treated cells (40 or 160 µg/mL). 

In order to distinguish between different possible scenarios, the clonogenic efficiency was also evaluated in cells treated with BLEO (40 µg/mL) or BLEO + OLA (50 nM) (Figure 1B). Two conclusions could be derived. First, taking into account the errors, cell viability results resembled clonogenic efficiencies (BLEO: 48 vs. 52%; BLEO + OLA: 41 vs. 50%), indicating that in the presence of 40 µg/mL BLEO, about one in every two cells was alive and cycling. Second, upon the OLA treatment, no difference was observed.

Although an even lower OLA concentration (25 nM) is known to have effects on VERO nuclear PARP-1 activity [43], and 50 nM OLA is enough to prevent or partially revert the epithelial-to-mesenchymal transition induced by TGF-β in NMuMG cells [52], a higher OLA concentration was assayed as well, just in case an unexpected shift occurred. As can be seen in Figure A2, the OLA concentration was tripled (to 150 nM) and still displayed no effect on the BLEO-treated cells.

OLA did not potentiate a BLEO lethal effect in VERO cells. The absence of potentiation of the BLEO effect was also evidenced with chemically different, less specific PARPis and with a PARG inhibitor, indicating that PAR metabolism was not crucially involved in the BLEO-induced DDR. The inhibitors used were 3-aminobenzamide (3AB), 5′-deoxy-5′-[4-[2-[(2,3-dihydro-1oxo-1H-isoindol-4-yl)amino]-2-oxoethyl]-1-piperazinyl]-5′-oxoadenosine dihydrochloride (EB), and 6,9-diamino-2-ethoxyacridine-DL-lactate monohydrate (DEA). Figure A3A represents PAR, its synthesis by PARPs, its degradation mainly by poly-ADP-glycohydrolase (PARG), and the inhibitors abbreviations associated with their targets. Figure A3B depicts the PAR quantification on the control untreated cells and cells treated with PARPis or the PARG inhibitor DEA. As the basal PAR was low and this was done once, these measurements did not have much sensitivity, but overall, they were a control to check that the inhibitors were active. The lack of potentiation [25] of BLEO effects by PARPis 3AB or EB was demonstrated (Figure A3C,D). Finally, PARG inhibition did not change the cell viability in the presence of BLEO (Figure A3E,F).

To sum up, despite being able to alter the PAR metabolism, neither PARP nor PARG inhibitors potentiated the toxic effects of BLEO in VERO cells. 

### 2.2. Untreated VERO Cell Nuclei Harbor PARP, PARG, and PAR

Next, it was checked whether VERO cells were expressing some of the nuclear molecular actors of PARylation, as well as synthesizing basal PAR. As displayed in Figure 2A–D, the indirect immunocytofluorescence (ICF) and DAPI (blue) counterstain showed that nuclear PARP-1/2 (green) was distributed throughout the nucleus, while the PARG (red) distribution was punctuated and excluded the nucleolus. Relative intensity measurements (Figure 2E,F) following the lines drawn in Figure 2A,B, respectively (color-coded like the channels), also supported these observations. Regardless of the distribution, the important point is that VERO cells were expressing at least PARP-1/2 and PARG. Basal PAR was also detected, as demonstrated by the comparison of Figure 2H vs. Figure 2K and the respective relative intensity graphs (Figure 2L,M).

### 2.3. No Sharp PAR Increase Could Be Detected Immediately after the 45 min Pulse of BLEO

The first estimations in the literature suggested that PAR can increase up to 500-fold in response to a genotoxic insult [2]. Later, 50-fold increases under PARG inhibition and 7-fold increases on specific proteins have been reported [2,3]. To assess whether VERO cells respond to BLEO by increasing PAR levels, we performed inmunocytofluorescence experiments immediately after the end of the treatment (t = 0) with three different anti-PAR antibodies.

The 10H anti-PAR antibody has a known specificity for long PAR chains (above 20 residues) [53] and has been widely used to monitor the nuclear response to DNA damage [24,54]. Interestingly, some DDR proteins do not interact with short PAR chains (16-mer), while long PAR chains (55-mer) promote their integration into protein complexes [55]. Thus, the best antibody for detecting long-chain PAR induced by genotoxic stress is 10H anti-PAR. In DAPI-counterstained control or BLEO-treated (40 µg/mL, 45 min) cells (blue, Figure 3A–C), one in every several cells in the population displayed a strong nuclear PARylation signal, while the rest displayed no signal (Figure 3E; in Figure 3F, the red point under the calibration bar is a single H10-anti-PAR-positive nucleus). This observation would explain why the PAR increase in the cell population was not significant (Figure 3I, right-hand graph). As can be seen in Figure A4, a different cell type used as a positive control (CHO9 fibroblastic cell line) displayed nuclear PARylation more frequently in the same experimental conditions.

It has formerly been reported that nuclear PAR in untreated VERO cells is detected with polyclonal rabbit BD anti-PAR or chicken Tulip anti-PAR antibodies but not with Tulip monoclonal 10H clone antibodies [57]. Thus, now we are reporting that a third antibody detected basal nuclear PAR in VERO cells. BD and ENZO anti-PAR antibodies are better suited to detecting short-chain PAR. According to the manufacturers, ENZO anti-PAR (BML-SA216) was specifically designed against “purified poly(ADP-ribose) polymer (chain length of 2–50 units).” This would explain why no signal was detected in the control cells with 10H anti-PAR (Figure 3D) but there was a PAR signal in control cells with BD (Figure 3G) or ENZO anti-PAR (Figure 3M).

No signal increase was detected with BD anti-PAR in BLEO-treated cells (Figure 3H, extracted from unpublished [56]). As a positive control, a slight signal increase (about ×1.5) was detected with BD anti-PAR only under extreme conditions leading to cell death (Figure A4). More recently, the anti-PAR reagent MABE1031 was also used to evaluate the PAR increase in response to H_2_O_2_, which was hampered in the presence of 50 nM OLA (Figure A4I).

Compared to control samples (Figure 3J,M), a slight but significant PAR increase (Figure 3I) was detected with ENZO anti-PAR due to the BLEO treatment (Figure 3K,N). Of notice, given the nature of BLEO treatment, some cells may have been damaged 45 min before and other ones just at the very last minute before fixation. Therefore, in this single fixation, we may have had many time points that were superimposed. Regarding the effect of OLA in BLEO-treated cells (Figure 3L,O), there was a tendency toward PAR diminution. In fact, unlike what happened in BLEO-treated cells, in BLEO + OLA treated cells, the PAR level was indistinguishable from the control (*p* = 0.589). Therefore, the next step was to check that OLA was not interfering with the DNA damage induction by BLEO.

### 2.4. OLA Did Not Hamper the DNA Damage Induction by BLEO

DNA damage induction was registered in the DAPI-counterstained cells (blue, Figure 4A–C) through γH2AX (red, Figure 4D–F) and 53BP1 (green, Figure 4G–I) detection using ICF, as well as in cells subjected to single-cell gel electrophoresis or a comet assay (Figure 4I–K). The percentage of cells with a γH2AX foci (Figure 4M), the percentage of cells with a pan-nuclear γH2AX signal (Figure 4N), and the relative DNA damage index (DDI, Figure 4O) were quantified from three independent experiments. Furthermore, in one of the experiments, the relative number of γH2AX, 53BP1, and mixed foci per cell was also evaluated (Figure A5). All our DNA damage induction measurements had the same stair-shape, from left to right: the control in the lower step, then BLEO, and then BLEO + OLA in the higher step. As this happened regarding the percentage of cells with γH2AX foci (Figure 4M), the percentage of pan-nuclear γH2AX cells (Figure 4N), the comet DDI relative to the control (Figure 4O), and even the relative abundance of γH2AX, 53BP1, and mixed foci (Figure A5), it was concluded that at t = 0, the BLEO cells were more damaged than the control cells. Moreover, the OLA undoubtedly did not interfere with BLEO to avoid the initial DNA damage induction.

### 2.5. OLA Did Not Potentiate BLEO, Even in the Presence of a Pool of DNA Repair Enzyme Inhibitors

All the data above suggested that VERO cells were repairing DSBs induced by BLEO via a mechanism that was independent of PARPs-1/2 and 3 since OLA can block their enzymatic activity or induce enzyme trapping. According to some authors [8,58], PARP-1 activation participates in MMEJ but not on HR or c-NHEJ. If this was the case, then the simultaneous block of HR and c-NHEJ should leave the cells dependent on MMEJ, thus dependent on PARP-1. Therefore, we hypothesized that the simultaneous inhibition of DNA-PK, LigIV, and ATM (3i) would either induce the death of remnant cells or force them to shift to MMEJ, thus becoming PARP-dependent and OLA-sensitive.

As depicted in the Figure 5 legend, 3i was toxic per se (viability = 66.82%) and BLEO + DMSO viability was 41.46%. According to Webb’s equation, the expected viability if the combined effect of BLEO and 3i was additive was 27.70%. The real viability was 26.42 ± 1.25%. Thus, the 3i toxic effect was added to the BLEO effect. 3i did not potentiate the BLEO effect. 

To sum up, the unexpected results were that neither 3i nor OLA potentiated BLEO effects on the cell viability and that most (≈75%) of the cell subpopulation that resisted BLEO was still resistant in the presence of 3i and OLA. 

## 3. Discussion

Resistance to BLEO itself in VERO cells can be due to several causes, including a low intake, high export, non-availability of free radicals (due to low Fe, hypoxia, or high antioxidant defenses) [59] or BLEO inactivation by BLEO hydrolase [60]. From these known putative causes, hypoxia was discarded since experiments were carried out under aerobic conditions. Furthermore, as BLEO sensitivity does not increase in the presence of the BLEO hydrolase inhibitor E-64, it can be inferred that the VERO cells did not express significant levels of this inactivating enzyme.

VERO cells have low basal PAR levels, which increase moderately in response to different genotoxic insults. Having tried three different anti-PAR antibodies, we did not detect a sharp increase in PAR levels in response to BLEO. ICF with a 10H antibody for long-chain PAR showed no basal long-chain PAR and a strong signal in very few BLEO-treated cells was observed. Although it is known that PAR levels can peak and then diminish, the nature of the BLEO treatment implied that at the fixing time, the damage induced at the beginning of the pulse coexisted with the damage induced at the end of the pulse (45 min later). Thus, if VERO cells had been good PAR synthesizers in response to BLEO, we should have detected different degrees of the PARylation signal regardless of its kinetics. PARylation in response to other agents used as positive controls, such as H_2_O_2_ or MMS, was also modest (a very high concentration of the latter, which induced cell death, was necessary to achieve a clear PAR increase). The VERO cells expressed PARP-1/2 and PARG, which were clearly detected by the ICF (Figure 2). Recent evidence shows that a low nuclear PARP-1 level is not the reason why VERO cells are not good PAR synthesizers. According to published Western Blots (WB) results, the level of PARP-1 expression in VERO cells is considerable, with it being similar to that in human epithelial cell lines, such as HEK293 (embryonic kidney), A549 (lung carcinoma) [61], or HEp-2 (laryngeal carcinoma) [62]. In fact, VERO has been chosen as one of the model epithelial cell lines for analyzing the effects of PARP-1 knockdown [61]. The VERO PARP-1 protein sequence (XP 007986456.1) harbors several point amino acid substitutions despite showing a good alignment with hPARP-1 (NP 001609.2). An effect of such mutations in PARP-1 structure and/or activity cannot be formally discarded since several hPARP-1 mutations causing diminished enzymatic activity have been identified [63] 

The lack of the potentiation of BLEO-induced damage by OLA was not surprising given the cumulative evidence indicating that OLA can display a range of effects from pro-cell death to protective effects depending on the cellular context [11,13,14,15,16,17,18,19,20,21,22,23,24,28,29]. OLA promotes cell death differentially in cells that have hampered DSB repair (due to a BRCA mutation for example) rather than in cells that have a normal, fully functional, DNA repair systems. As VERO cells do not harbor known mutations in DNA repair genes, this was an expected result. However, the initial DNA damage was measured (γH2AX positive cells, γH2AX and 53BP1 foci number, and γH2AX pan-nuclear cells and DNA damage index) to exclude the possibility that OLA had interfered with the DNA damage induction. It was also confirmed that the 50 nM OLA treatment reached its targets and affected the PARylation. ENZO anti-PAR antibody, which was generated against a 50-unit PAR chain, depicted a measurable and significant PAR increase in the presence of BLEO, which decreased, as expected, under co-treatment with 50 nM OLA (Figure 3). Finally, the WB confirmed that the PARylation increase induced by H_2_O_2_ (detected using MABE1031 anti-PAR reagent) was inhibited by a co-treatment with 50 nM OLA (Figure A4, panel I). An interesting point is that, according to our preliminary data (Figure A4), OLA increased the 53BP1 foci number. These data regarding VERO cells agree with data in HeLa and PtK2 cell lines published by Saquilabon et al. [64], who determined that PARPis induce the increased recruitment of 53BP1 to DNA damage sites. It has been interpreted that PARP-1/2 activation is antagonistic to c-NHEJ, limiting the 53BP1 accumulation at damaged sites [64]. Antagonistic roles in the selection of DSB DNA repair pathways have been documented for PARP-1 vs. Ku [58,65] and BRCA vs. 53BP1 [66,67].

The lack of abundant PAR synthesis in response to DNA damage induction and the absence of potentiation by OLA could be due to the fact that VERO cells were repairing BLEO-induced DNA damage, mainly through PARP-independent pathways. In such a case, blocking these pathways would force cells to become PARP-dependent or die. For this reason, and as an effort to help delineate the boundaries of synthetic lethality, a pool of inhibitors (3i) targeting key enzymes of the two main pathways of DSB repair, namely, HR (ATM inhibitor) and C-NHEJ (DNA-PK and LigIV inhibitor), were used. Cells were exposed to BLEO and a co- and post-treatment with 3i alone or 3i + OLA. 

The most important point was that since in VERO cells we did inhibit ATM, DNA-PK, and LigIV, and under that condition, a cell subpopulation survived, even in the presence of the PARP inhibitor OLA, no synergism occurred. In fact, our results indicated the presence of a VERO subpopulation that was resistant to BLEO, BLEO + OLA, and BLEO + OLA + ATM inhibitor KU55933 + DNA-PK inhibitor KU-0060648 + LigIV inhibitor SCR7 pyrazine (Figure 5). 

An improved understanding of the synthetic lethality strategies will probably extend the use of PARP inhibitors beyond tumors with BRCA1/2 mutations [68]. Aside from drug–drug interactions, known causes for OLA resistance include disturbed PARylation metabolism, alterations in drug transporters, up-regulation of HR, and replication forks [69,70,71].

As discussed above, although the VERO cells expressed normal PARP levels, poor PARylation was observed and VERO cells harbored PARP mutations of unknown biological meaning. This deserves further study. PARPis resistance has been described in an ovarian cancer patient carrying a point mutation in PARP-1 [70]. Regarding transporters, it is known that P-glycoprotein (P-gp) transporters export OLA, pumping it out of the cell [41]. We have not evaluated whether VERO cells overexpress P-gp transporters. However, PAR diminution in the presence of OLA evidences that OLA is reaching PARP, making P-gp overexpression an unlikely explanation of OLA resistance in VERO cells. 

Assuming that all the used inhibitors worked as in other models, one explanation regarding the resistance to BLEO + OLA + 3i would be that cells repaired DNA damage through a PARP-independent pathway rather than MMEJ (which is PARP-1 dependent) or HR (because ATM was inhibited) or c-NHEJ (since DNA-PK and LigIV were inhibited). Even under the HR defect (due to mutated or inhibited proteins), replication fork stabilization has been proposed as an alternative PARPis resistance mechanism [69,71]. According to Haynes et al. [72], checkpoint kinases ATR, CHK1, and WEE1 play different roles in replication fork stabilization, providing alternative mechanisms to be considered in combination therapy development in order to avoid drug resistance. For example, ATRis (VE-821, VE-822, or AZ20) overcome acquired and pre-existing PARPis resistance in multiple BRCA1-deficient cancer cell lines of distinct origins. Moreover, such cells are preferentially affected over BRCA-proficient or PARP-sensitive cells [73]. ATRi (AZD6738) and WEE1i (AZD1775) display differential activity in a subpopulation of non-germinal-center-B cell (non-GCB) diffuse large B-cell lymphoma (DLBCL) cell lines characterized by high MYC oncogene expression and cyclin-dependent kinase inhibitor 2A/B (CDNK2A/B) deletion [74]. Interestingly, CDKN2A codes alternatively spliced variants that are involved in cell cycle control, including the tumor suppressor p16INK4a. In turn, CHK1 inhibitor Prexasertib is effective against human head and neck squamous cell carcinoma cell lines with *CDKN2A* genetic losses. Conversely, restoration of p16 expression in hypersensitive cells prevents Prexasertib-induced cell proliferation drop [75].

Does the DDR response and coordination of DNA repair and cell cycling work properly in VERO cells? VERO cells have a homozygous ≈9 Mb deletion on chromosome 12, causing the loss of CDKN2A/B genes, besides the type I interferon gene cluster (which probably explains why they are so infection-prone). Although these mutations by themselves are not enough to transform cells into tumorigenic cells, the loss of CDKN2A/B may play a crucial role in the acquirement of immortality in the VERO cell lineage [49]. Interestingly, *CDKN2A* gene mutations are found in up to 40% of familial cases of melanoma, up to 25% of head and neck squamous cell carcinomas (HNSCC), some breast cancers and pancreatic cancers, and others. In some families, *CDKN2A* gene mutations facilitate the development of only one type of cancer, while in other families, they can lead to a cancer predisposition syndrome, increasing the risk of developing multiple types of cancer. Furthermore, somatic *CDKN2A* gene mutations have been found in some people with brain tumors and in children with acute lymphoblastic leukemia [76] (From this point of view, VERO cells could be considered a cancer-prone cell model. Furthermore, the effects of mono-treatments and combined treatments with checkpoint kinases inhibitors should be tested. Moreover, further research is needed to know whether CDKN2 mutations are mandatory over mutations regarding its synergism with PARPis. For example, does a BRCA mutated cell lose its synergism with PARPis in general, and with OLA in particular, if it is also mutated in *CDKN2A*? Could this be an explanation of why some BRCA-mutated breast cancers do not respond to OLA? Efforts to determine the synthetic lethality partners of PARPis [68], ATRis, CHK1is, and WEE1is are being done. Both positive and negative results will be useful for understanding the underlying mechanisms.

## 4. Materials and Methods

### 4.1. VERO Cell Culture

*Cercopithecus aethiops* (green monkey) VERO cells (ATCC CCL-81, [57]) were cultured in MEM-STA (Capricorn, Capricorn Scientific GmbH, 35085 Ebsdorfergrund, Germany) supplemented with 10% FBS (Capricorn # FBS-11A, collected in South America), penicillin/streptomycin (Capricorn PS-B, 100×) and 2 mM L-glutamine at 37 °C and 5% CO_2_. Cells were seeded onto 96-well cell culture plates for MTT assay, round coverslips in 24-well plates for immunocytofluorescence (ICF), 30 mm dishes for the comet assay, and 50 mm dishes for the clonogenic assay.

### 4.2. Treatments with BLEO and Inhibitors

Bleomycin sulfate treatment (BLEO); from NOLVER (Montevideo, Uruguay), NIPPON KAYAKU (Japan), or LKM (Peru) consisted of a 45 min pulse in the absence of FBS. Controls and other experimental conditions were also subjected to 45 min FBS depletion. Initially, dose–response curves (0, 4, 10, 20, 40, 80, 160, 200, and 500 μg/mL) were found. The Olaparib (Tocris, Minneapolis, MN, USA) treatment was continuous. Dose–response curves (0, 50, 100, 150, and 200 nM) were also found. Then, 40 μg/mL BLEO and 50 nM OLA were selected for the following experiments: OLA effect on the PAR pool was checked and compared with other PARPis and with the PARG inhibitor DEA in untreated VERO cells using ICF with a BD anti-PAR antibody (Figure A3B). While DEA increased PAR, 25 nM OLA did induce an effect that was very similar to 250 nM OLA, diminishing the endogenous PAR by more than 100 nM EB or 5 mM 3AB. Like with OLA, the treatment with ATM, DNA-PK, and Lig IV inhibitors, namely, 10 μM KU55933 (SIGMA SML-1109, St. Louis, Missouri, USA [77]), 0.1 μM KU-0060648 (SIGMA SML-1257, St. Louis, MO, USA), and 0.1 μM SCR7 Pyrazine (SIGMA SML-1546, St. Louis, MO, USA [78]), was continuous. SIGMA is subsidiary of Merck KGaA (Darmstadt, Germany), and its headquarter is at St. Louis, Missouri, USA.Co-treatments were done with the correspondent inhibitors + BLEO for 45 min; then, for the MTT or clonogenic experiments, BLEO was removed and the inhibitors were added again in a fresh medium with FBS. For the comet assay or ICF, the experiment was stopped and the cells were fixed or lysed immediately after the 45 min treatment. All times in the graphs refer to the post-BLEO-treatment time. As such, t = ”0” means “immediately after the 45 min BLEO treatment”, t = 24 is 24 h later, and so on.

### 4.3. Verification of the PARPis and PARGi Effects on VERO Basal PAR Pool

ICF using an anti-PAR antibody on untreated VERO cells indicated that 25 nM OLA had an effect that was very similar to 250 nM OLA, diminishing the endogenous PAR more than 100 nM EB or 5 mM 3AB. Conversely, the PARG inhibitor DEA induced PAR accumulation.

### 4.4. Cell Viability Assay (MTT)

Cells were seeded in 96-well plates at a density according to the duration of the experiment (e.g., 15,000/well for 24 h experiments and 3000/well for 72 h experiments). The cell viability was determined by a 3-(4,5-dimethylthiazol-2-yl)-2,5-diphenyltetrazolium bromide (MTT) colorimetric assay [79], in which metabolically active cells reduced the dye to purple formazan. Cells were incubated for 1 h at 37 °C with MTT (0.5 mg/mL final concentration in 10 mM glucose in PBS). Formazan crystals were dissolved with DMSO. The absorbance was measured using a reference wavelength of 630 nm and a test wavelength of 570 nm on a Varioskan Flash microplate reader, Thermo Scientific, Waltham, MA, USA).

### 4.5. Clonogenic Assay

Cells were seeded in 50 mm cell culture dishes at a very low density (1000 cells for treatments and less (500 or 250) for the controls). After the attachment, cell cultures were treated for 45 min with BLEO in the absence or presence of inhibitors. Then, the culture media was changed and fresh inhibitors were added. The cells were incubated until control colonies reached at least 50 cells (counted under the microscope) for 8 days. Finally, colonies were fixed on 70% ethanol at room temperature (RT) for 10 min, briefly stained with 0.1% crystal violet (Fluka 61135), rinsed in abundant distilled water, dried at 37 °C, and manually counted by an observer under a magnifying glass. The plates were blind-coded. The clonogenic efficiency was expressed relative to the controls.

### 4.6. Comet Assay

The assay was carried out in alkaline conditions to detect SSB, DSB, and alkali-labile sites [80,81,82]. The slide’s surface was pretreated with 1% normal melting point agarose (Sigma, St. Louis, MO, USA) in PBS. After a 45 min treatment, the cells were washed with PBS, incubated with trypsin-EDTA for 5 min at 37 °C, centrifuged, and resuspended in PBS; then, 20 µL of cell suspension was mixed with 80 µL of 0.75% low-melting agarose (Sigma, St. Louis, MO, USA) in PBS at 37 °C. Immediately, an 80 μL volume of the suspension was placed on a slide, covered with parafilm, and kept at 4 °C (10 min). The parafilm was removed and the slide was immersed in a cold lysis buffer (2.5 M NaCl, 100 mM EDTA, 10 mM Tris-HCl, and 8 g of NaOH/890 mL of water, adjusted to pH 10, to which was added 1% Triton-X-100 and 10% DMSO an hour before use) and kept at 4 °C (from 1 to 15 days). The slides were incubated in cold electrophoresis buffer (300 mM NaOH and 1 mM EDTA at pH 13) for 20 min at 4 °C to unwind the DNA strands and expose the alkali-labile sites. Electrophoresis was performed at 25 V for 20 min. The buffer volume was adjusted to achieve a current intensity in the range of 250 to 300 mA. After that, the slides were washed with neutralization buffer (0.4 M Tris-HCl, pH 7.5) three times for 5 min each and washed with distilled water. Subsequently, the slides were stained with 80 μL of DAPI (6 μg/mL) (10 min), washed with distilled water, and covered with coverslips.

Comets were blind-counted under epifluorescence and representative photographs were taken with the confocal microscope in non-confocal conditions (confocal aperture 5) under a 20× objective and classified according to the degree of damage in five categories: from 1 to 5 (*α* = degree of damage). The following criteria were used to assess the degree of damage: 1: no damage and intact or with a halo surrounding the core; 2: a little damage (the DNA was distributed in an oval); 3: the anterior–posterior axis measured twice the diameter; 4: the compact DNA was reduced and a large cloud of DNA (long tail of the comet) appeared; and 5: the tail was separated from the rest of compact DNA.

A double-blind count of 100 cells was performed. A damage index (DDI) was calculated using the equation: DDI = Σ(n.α)P, where *α* (which can range from 1 to 5) expresses the degree of damage and *n* is the number of cells with the degree of damage *α*. 

### 4.7. Indirect Inmunocytofluorescence and Image Acquisition

Cells were washed with filtered PBS (fPBS, 0.22 µm pore size), fixed in 4% paraformaldehyde (PFA) in fPBS 15 min at 4 °C, washed in fPBS, permeabilized in 0.1% Triton-X100 in fPBS, and immersed in blocking buffer (0.2% Tween, 1% BSA in fPBS) for 30 min. Briefly, cells were incubated with the specific antibodies, namely, 1:300 rabbit anti-PARP-1/2 (Santa Cruz sc-7150, CA, USA), 1:500 rabbit anti-PAR (BD551813, Becton Dickinson (Franklin Lakes, NJ, USA)), 1:50 mouse anti-PAR (Enzo BML-SA216, Farmingdale, NY, USA), 1:100 10H-anti-PAR (Tulip #1020), 1:200 anti-hPARG (Abcam 16060, Cambridge, MA, USA), 1:400 mouse monoclonal anti-γH2AX (Abcam), and 1:300 anti-53BP1 (Abcam 36828), and diluted in blocking buffer for 2 h at 37 °C. After washing in fPBS/T (0.1% Tween), cells were incubated (1 h, RT) with the correspondent anti-antibodies mix (1:250 anti-mouse-Cy3 Jackson Immuno Research, 1:1000 goat-anti-rabbit 488 (#A-11034, Thermo Fisher Scientific, Waltham, MA, USA) in a blocking buffer. After washing in fPBS/T and fPBS, DAPI counterstaining (1.5 µg/mL in fPBS), and a final wash in fPBS, the coverslips were mounted in Prolong Gold (Molecular Probes P36930, Eugene, OR, USA) and sealed with nail polish. Controls without a primary antibody were run in parallel to check the specificity of the signals.

Mainly an Olympus BX61/FV300 (Tokyo, Japan) and sometimes a Zeiss LSM800-Airyscan (Oberkochen, Germany) or a Leica microscope were used to take the confocal images/stacks. Fluorescence excitation was performed with the following lasers: diode 405 nm (DAPI), multiline argon 488 nm (Alexa Fluor 488), and helium-neon 561 nm (Cy3). Microscope settings were adjusted to register no signal in controls without primary antibodies. The scanning of optical sections was performed sequentially for the different fluorochromes. All images in each experimental series were taken with the same setting at the same confocal session. If modified, all were subject to the same degree of brightness/contrast adjustment and Gaussian blur filtering, including the control without a primary antibody. ImageJ free software [83] was used for the image processing.

### 4.8. Cell Counting Using Low-Magnification Fields

The ImageJ “Cell Counter” plug-in [84] (was employed to find out the percentage of nuclei that exhibited no γH2AX, well-defined γH2AX foci, or γH2AX pan-nuclear staining.

### 4.9. Relative PAR Quantification

The total field PAR intensity (RawIntDen, the sum of signal intensity in all the image pixels) was normalized using DAPI RawIntDen in each field (to account for putative differences in cell densities). Then, the data were expressed as a percentage of the control.

### 4.10. Statistical Analysis

The results were expressed as mean ± SEM. Differences between different experimental groups were tested for significance using a two-tailed unequal variances Student’s *t*-test in Microsoft Excel 2007 [85] or a one-way analysis of variance (ANOVA), followed by post-hoc multiple comparisons tests (Tukey, Scheffe, Bonferroni, and Holm) in ASTATSA [86])

## Figures and Tables

**Figure 1 ijms-21-08288-f001:**
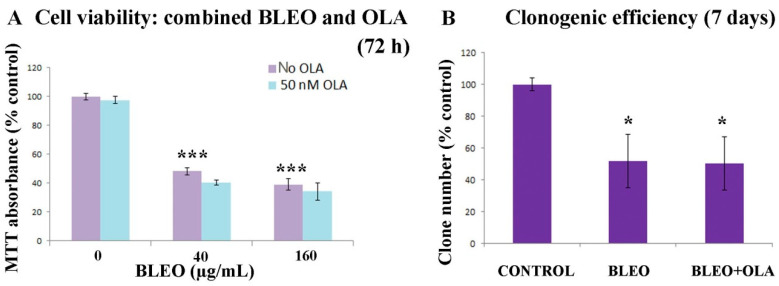
Neither the cell viability loss nor clonogenic efficiency loss induced by BLEO in VERO cells were potentiated by OLA. (**A**) Cell viability (MTT assay). Cells were exposed to BLEO and, when indicated, co- and post-exposed to 50 nM OLA. The result was evaluated after 72 h. Most experiments were carried out using 40 µg/mL BLEO and 50 nM OLA. The respective *n*’s were as follows. No OLA: 63, 72, and 22; 50 nM OLA: 75, 86, and 12 (see Figure A2 for 150 nM OLA). ANOVA (*p* = 1.11 × 10^−16^). Post-hoc tests: Tukey, Scheffe, and Bonferroni. ***: *p* < 0.001. (**B**) Clonogenic efficiency of VERO cells in the control condition or under a pulse treatment with 40 µg/mL BLEO (45 min) in the absence or presence of continuous treatment with 50 nM OLA. Data were from two independent experiments in triplicate. All results are expressed as mean ± SEM. Comparisons against control. ANOVA (*p* = 0.0371) and Holm *p*-value with only comparisons against the control considered. *: *p* < 0.05.

**Figure 2 ijms-21-08288-f002:**
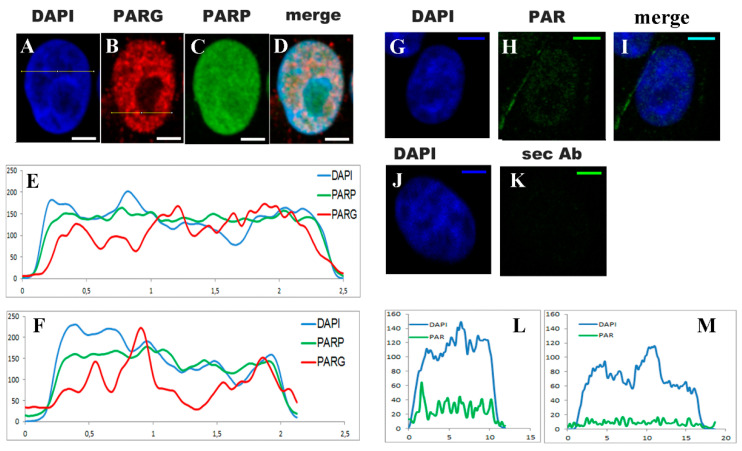
PAR, PARP, and PARG were detected in the VERO cell nuclei. (**A**–**D**) DAPI (blue), PARG (red), PARP (green), and the merged confocal images of representative nuclei. (**E**,**F**) Graphs displaying the fluorescence intensity measurements in the three channels of the correspondent nuclei images through two lines that are drawn in (**A**) or (**B**) respectively. Intensity in Relative units. Distance: 1 U ≈ 5 µm (**G**–**I**) Indirect immunocytofluorescence (ICF) with BD anti-PAR antibody. DAPI (blue), PAR (green), and merged channels. **(J**,**K**) Control of the anti-PAR ICF without the anti-PAR antibody with only the secondary antibody (sec Ab). (**L**,**M**) Blue and green channel intensities measured over a line in (**H**) (with anti-PAR) and (**I**) (without anti-PAR), respectively. Confocal images were obtained with the same settings and subject to identical processing adjustments. Relative intensity on the ordinates and distance in µm on the abscissas. PAR signal (green) was low but detectable in untreated VERO nuclei. Bar: 5 µm.

**Figure 3 ijms-21-08288-f003:**
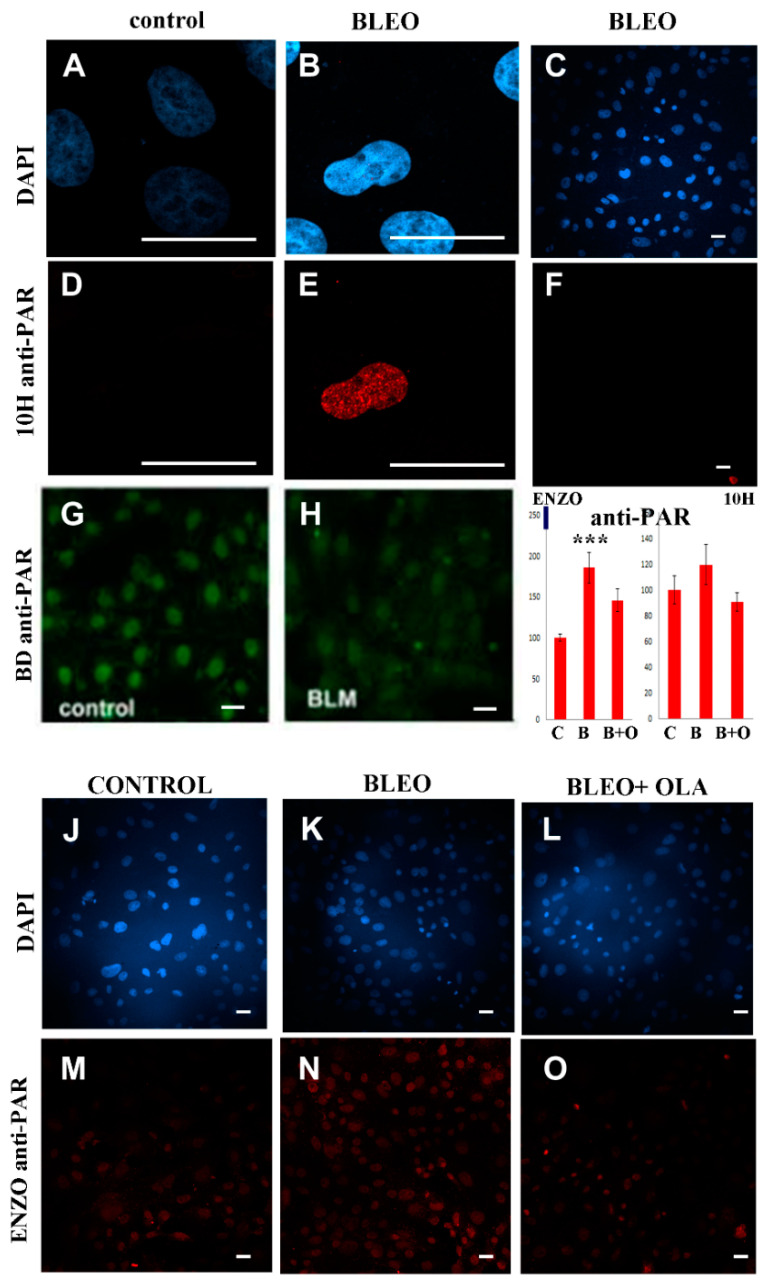
Neither of the three anti-PAR antibodies evidenced a sharp PAR increase using ICF immediately after the BLEO (45 min) treatment. This was observed in at least five independent experiments. (**A**–**C**) DAPI channel to see nuclei that were positive or negative for the (**D**–**F**) 10H anti-PAR signal. (**A**,**D**) control and (**B**–**F**) BLEO. (**G**,**H**) BD anti-PAR in (**G**) control or (**H**) BLEO-treated cells. (**I**) To estimate the signal increase, the PAR signal intensity was quantified in one of the experiments with two of the antibodies. The whole-field PAR intensity was adjusted by the DAPI intensity and expressed as a percentage of the control. Mean ± SEM. Left: ENZO anti-PAR. ANOVA (*p* = 0.0010). Post-hoc test: Holm against control, Tukey, or Scheffé. ***: *p* <0.001. Right: 10H anti-PAR antibody showed no differences. (**J**,**M**) Control, (**K**,**N**) BLEO-, or (**L**,**O**) BLEO + OLA-treated cells. Panels **G** and **H** were extracted from Lafon-Hughes’ unpublished Ph.D. thesis [56]. Bar: 25 µm.

**Figure 4 ijms-21-08288-f004:**
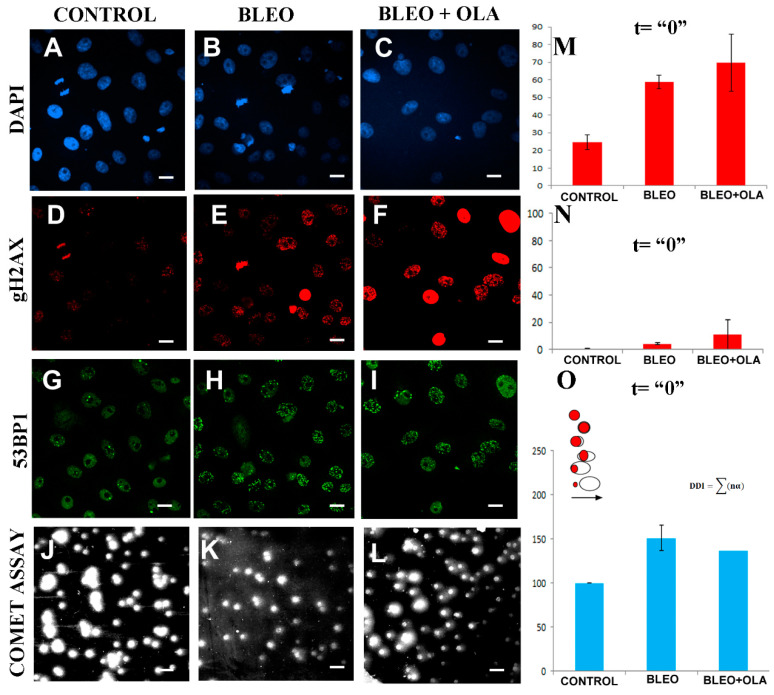
OLA did not hamper the DNA damage induction by BLEO. The initial (t = ”0”) BLEO-induced DNA damage in the presence or absence of the OLA co-treatment was evaluated All images and graphs follow the order: Control, BLEO and BLEO+OLA, from left to right. (**A**–**I**) Representative ICF images. (**A**–**C**): DAPI in blue; (**D**–**F**): gH2AX in red; (**G**–**I**): 53BP1 in green. (**J**–**L**) Representative single-cell electrophoresis or comet assay images (**M,** in red): Percentage of cells with a γH2AX foci. Data were from three independent experiments and *n* ≥ 200 cells/condition. Two-tailed *t*-test. (**N,** in red) Pan-nuclear γH2AX cells were counted on the same set of images. (**O**, in blue) The comet assay was evaluated using the DNA damage index (DDI). In turn, the results were normalized to control the DDI to pool the experiments. Data were from three independent experiments, with *n* ≥ 140 cells/condition. Two-tailed *t*-test. *p* = 0.17 (control vs. BLEO), *p* = 0.099 (control vs. BLEO + OLA), *p* = 0.24 (BLEO vs. BLEO + OLA). To calculate the DDI, each comet was assigned a degree from 1 to 5. The inset scheme represents comet heads (in red) and comet tails (in white) for successive degrees. 1: compact or with a simetric halo (first two drawings); 2: asimetric halo; 3: tail length = head length; 4: tail length > head length; 5: head separated from tail. Bar: 20 µm.

**Figure 5 ijms-21-08288-f005:**
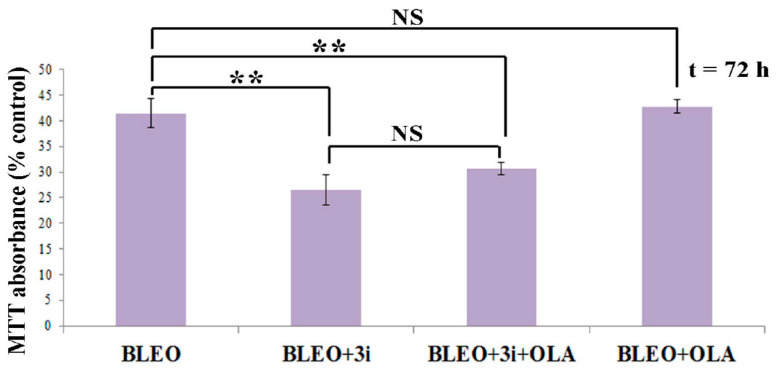
OLA did not potentiate the BLEO effect in VERO cells, even in the presence of certain inhibitors of DSB repair pathways. The cell viability relative to the control (MTT, 72 h) of BLEO-treated cells in the absence or presence of OLA and “3i” (0.1 µM DNA-PK inhibitor KU-0060648 + 0.1 µM LigIV inhibitor SCR7 pyrazine + 5 mM ATM inhibitor KU55933). DMSO, the inhibitors’ vehicle, was maintained in a constant concentration in all conditions. Data were from six independent experiments, with *n* = 32, 56, 54, and 54. Mean ± SEM. According to ANOVA and Tukey, Schaffé, or Bonferroni a posteriori tests, OLA made no difference, whereas 3i or 3i + OLA differed from the BLEO treatment with **: *p* < 0.01. NS: statistical non-significance. Although not represented in the graph, 3i was toxic by itself (cell viability = 66.82 ± 2.9%).

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
