# Peer review of "PARP Inhibitor Olaparib Causes No Potentiation of the Bleomycin Effect in VERO Cells, Even in the Presence of Pooled ATM, DNA-PK, and LigIV Inhibitors"

_ijms, 2020, doi:10.3390/ijms21218288_

Round 1
Reviewer 1 Report
In this manuscript, Perini et al describe that the PARP inhibitor olaparib has no effect on enhancing the cytotoxic effect of bleomycin on VERO cells. It is an observational study in which no underlying mechanism is provided. The authors' conclusion that neither the use of the 3 inhibitors (DNA-PK inhibitor, LigIV inhibitor and ATM inhibitor) nor olaparib enhances the effects of bleomycin treatment on cell viability does not seem to correspond with the data in figure 5.
Minor points
Line 18: “PARP activation” instead of “PAR activation”
Author Response
REVIEWER 1
(x) I would not like to sign my review report
( ) I would like to sign my review report
English language and style
( ) Extensive editing of English language and style required
( ) Moderate English changes required
( ) English language and style are fine/minor spell check required
(x) I don't feel qualified to judge about the English language and style
Yes |
Can be improved |
Must be improved |
Not applicable |
|
Does the introduction provide sufficient background and include all relevant references? |
(x) |
( ) |
( ) |
( ) |
Is the research design appropriate? |
( ) |
( ) |
(x) |
( ) |
Are the methods adequately described? |
(x) |
( ) |
( ) |
( ) |
Are the results clearly presented? |
( ) |
( ) |
(x) |
( ) |
Are the conclusions supported by the results? |
( ) |
( ) |
(x) |
( ) |
Comments and Suggestions for Authors
In this manuscript, Perini et al describe that the PARP inhibitor olaparib has no effect on enhancing the cytotoxic effect of bleomycin on VERO cells. It is an observational study in which no underlying mechanism is provided.
The NIH National Cancer Institute defines an observational study as “A type of study in which individuals are observed or certain outcomes are measured. No attempt is made to affect the outcome (for example, no treatment is given)”. Therefore, taking into account this definition we have performed an experimental study to address the question whether Olaparib would potentiate DNA damage induced by Bleomycin in VERO cells. Although we did not provide with certainty the actual mechanisms behind our results, in this work we were able to discard some mechanisms, including: (1) endogenous bleomycin hydrolase activity; (2) nuclear PARP-1/2 depletion; (3) low olaparib uptake or high export by Pg-p leading to no olaparib effects; (4) olaparib direct interference with bleomycin; (5) low 53BP1 leading to c-NHEJ suppression, high homologous recombination and replication fork protection. Meanwhile, alternative hypothesis to explain these observations were proposed, including: (1) low PARP activity due either to some PARP mutation still not identified as detrimental or to upstream cell signals diminishing PARP activity; (2) the existence of a still not described, PARP-independent DNA repair pathway; (3) the loss of CDNK2 genes which is one of Vero cells characteristics. We think that all of this is worth being communicated.
The authors' conclusion that neither the use of the 3 inhibitors (DNA-PK inhibitor, LigIV inhibitor and ATM inhibitor) nor olaparib enhances the effects of bleomycin treatment on cell viability does not seem to correspond with the data in figure 5.
We apologize if data presented in Fig .5 led to some misunderstanding. It was not intended to address whether the 3 inhibitors enhanced the effects of bleomycin (as potentiation or as an additive effect). The important point was that olaparib did not enhance the effects of bleomycin alone nor of the combination of bleomycin plus the 3 inhibitors. In an effort to make the figure clearer to the reader, we have improved Figure 5 (line 298), adding bars to indicate what the asterisks refer to as well as the abbreviation “NS” to highlight statistical non-significance of comparisons involving olaparib (complementing the information in the figure legend, line 307)
Minor points
Line 18: “PARP activation” instead of “PAR activation”. Thank you. This typing error has been corrected.
Submission Date
30 September 2020
Date of this review

Reviewer 2 Report
Perini and collaborators have now revised their manuscript, answering to my previous requests:
As far as the level of PARP-1 in VERO cells, they now cited a recent work on J. Virol. The paper is identified only by DOI in the rebuttal letter (I find that it is cited as ref. 61). In this paper, there is no possibility for a direct comparison among different cell lines, since the blots referring to each of them has been not loaded on the same gel, or their exposure has been kept separately. That said, it appears that VERO cells contain PARP-1.
As far as the activity, the production of PAR and its inhibition by Olaparib, is now shown by supplementary Figure SD, panel I. The Authors try to explain these results by analyzing the sequence of PARP-1 in VERO cells and conclude that there is the possibility that some point mutation may affect PARP-1 enzymatic activity.
Minor points:
Please, acknowledge the use of MABE 1031 reagent in the appropriate section of Materials and Methods
Author Response
REVIEWER 2
06 Oct 2020 14:07:24
Open Review
(x) I would not like to sign my review report
( ) I would like to sign my review report
English language and style
( ) Extensive editing of English language and style required
( ) Moderate English changes required
(x) English language and style are fine/minor spell check required
( ) I don't feel qualified to judge about the English language and style
Yes |
Can be improved |
Must be improved |
Not applicable |
|
Does the introduction provide sufficient background and include all relevant references? |
(x) |
( ) |
( ) |
( ) |
Is the research design appropriate? |
(x) |
( ) |
( ) |
( ) |
Are the methods adequately described? |
( ) |
(x) |
( ) |
( ) |
Are the results clearly presented? |
(x) |
( ) |
( ) |
( ) |
Are the conclusions supported by the results? |
(x) |
( ) |
( ) |
( ) |
Comments and Suggestions for Authors
Perini and collaborators have now revised their manuscript, answering to my previous requests:
As far as the level of PARP-1 in VERO cells, they now cited a recent work on J. Virol. The paper is identified only by DOI in the rebuttal letter (I find that it is cited as ref. 61). In this paper, there is no possibility for a direct comparison among different cell lines, since the blots referring to each of them has been not loaded on the same gel, or their exposure has been kept separately. That said, it appears that VERO cells contain PARP-1.
The cited work (Xia et al., J Virol 2020, 94, e01572-19) corroborates our own data regarding PARP expression in Vero cells. Although we agree that for very precise comparisons, only samples loaded in the same gel would be accurate, we consider that the information presented, together with the citations, may help to clarify the reviewer´s doubt. Moreover, the use of an internal reference helps in the comparison across gels. In this case, GAPDH was selected as a reference and it can be seen that the ratio between PARP and GAPDH observed for Vero cells is similar to other cell lines. We believe this information strongly supports the evidence of PARP-1 expression levels in Vero cells.
As far as the activity, the production of PAR and its inhibition by Olaparib, is now shown by supplementary Figure SD, panel I. The Authors try to explain these results by analyzing the sequence of PARP-1 in VERO cells and conclude that there is the possibility that some point mutation may affect PARP-1 enzymatic activity.
The possibility that some point mutation may affect VERO PARP-1 enzymatic activity was addressed in previous versions of the manuscript (see crossed out lines 375-380) but we are happy to see that the now included figure helps clarify our suggestion. It has already been reported/pointed that VERO cells do have mutations in PARP, although none of them corresponds to the ones known as PARP-activity lowering mutations.
We also hope that the Western blot in Figure SD also helps demonstrate that Olaparib was effectively inhibiting PARP activity in Vero cells.
Minor points:
Please, acknowledge the use of MABE 1031 reagent in the appropriate section of Materials and Methods.
We thank the reviewer for this observation. We are willingly prepared to include this information in the main manuscript upon journal’s request. Meanwhile, methodological issues related to supplementary figures are included in legends.
Submission Date
30 September 2020
Date of this review
07 Oct 2020 12:13:43

Reviewer 3 Report
The authors of the ijms-967151 manuscript had already addressed the suggested corrections. I believe that the new data produced by the Western blot analysis are sufficient to consider that the manuscript deserves publication in Int. J. Mol. Sciences
Author Response
REVIEWER 3
(x) I would not like to sign my review report
( ) I would like to sign my review report
English language and style
( ) Extensive editing of English language and style required
( ) Moderate English changes required
(x) English language and style are fine/minor spell check required
( ) I don't feel qualified to judge about the English language and style
Yes |
Can be improved |
Must be improved |
Not applicable |
|
Does the introduction provide sufficient background and include all relevant references? |
(x) |
( ) |
( ) |
( ) |
Is the research design appropriate? |
(x) |
( ) |
( ) |
( ) |
Are the methods adequately described? |
(x) |
( ) |
( ) |
( ) |
Are the results clearly presented? |
(x) |
( ) |
( ) |
( ) |
Are the conclusions supported by the results? |
(x) |
( ) |
( ) |
( ) |
Comments and Suggestions for Authors
The authors of the ijms-967151 manuscript had already addressed the suggested corrections. I believe that the new data produced by the Western blot analysis are sufficient to consider that the manuscript deserves publication in Int. J. Mol. Sciences
We thank the reviewer for helping us to improve the manuscript throughout the previous revisions.
Submission Date
30 September 2020
Date of this review
06 Oct 2020 10:35:10

Round 2
Reviewer 1 Report
The authors have addressed my comments
This manuscript is a resubmission of an earlier submission. The following is a list of the peer review reports and author responses from that submission.
Round 1
Reviewer 1 Report
In this manuscript, Perini et al., describe that the PARP inhibitor olaparib has no effect on enhancing the cytotoxic effect of bleomycin on VERO cells. They finally postulate that CDKN2 deficiency may explain this process, although they do not conduct any experiment to prove it. I have the following comments:
1.- In the introduction, authors should be more precise in referring olaparib as a PARP inhibitor and not PARP-1 inhibitor.
2.- The reason for using the VERO cells is not clear. Perhaps the advantage of using these cells to study the effect of PARP-1 on the repair of bleomycin-induced DNA damage should be explained.
3.- Are VERO cells sensitive to olaparib in the absence of bleomycin?
4.- In Figure 2, which atibody has been used to detect PARP? Has PARP-1 or PARP-2 been detected? Perhaps the expression levels of PARP-1 and PARP-2 should be determined by western-blot.
5.- The authors should incorporate a positive control for PAR formation, either another agent that induces DNA damage or another cell lind that is known to induce PAR formation in response to DNA damage.
6.- Experiments should be included to validate the proposed hypothesis of the CDKN2 effect.
Reviewer 2 Report
In this manuscript, Perini and co-workers have analyzed whether Olaparib, a well known PARP inhibitor, may potentiate the killing effect of Bleomycin on VERO cells, in order to investigate the limits of synthetic lethality.
The problem of lack of effect of synthetic lethality approaches is important in cancer research for new chemotherapeutic regimens, as this may may vanish these efforts.
In their study, the Authors report that the cytotoxicity of Bleomycin reaches a saturation effect, suggesting the presence of a resistant subpopulation of VERO cells, and that Olaparib does not further potentiates the activity of the radiomimetic drug. In the attempt of better clarifying their results, the Authors have used DNA repair inhibitors, such as ATM, DNA-PK and DNA Ligase IV inhibitors, both in combination or not, with Olaparib. They show (Fig. 5) that these inhibitors were able to increase Bleomycin toxicity, while further addition of Olaparib did not, with no significant change in cell viability. They interpret these results as due to the presence of a subpopulation resistant to all these inhibitors and put forward some hypothesis in the Discussion.
Overall the results of the present study do not add new significant information in this field, since they have not investigated further the possible mechanisms of this phenomenon of drug resistance leading to absence of synthetic lethality. For instance, having suggested that VERO cells lack for CDKN2A gene, they have not substantiated this hypothesis with any data. RNA interference of CDKN2A in a suitable cell model would have added significant value to this manuscript. In addition, whether the lack of Olaparib effect is due to the presence of a resistant cell subpopulation has not been established. For instance, the limited toxicity could be due either to a cell subpopulation showing high drug export (reducing the amount of drug), or to a saturation effect of DNA damage induced by Bleomycin, whose action mechanism is dependent on cell metabolism, i.e. Fe and O2 (see ref. 18). Clarifying these aspects will further elucidate the mechanism of lack of synthetic lethality, which has not been addressed here.
Other points to be addressed:
Figure 1 and SA: a complete curve showing the dose-response at higher concentrations tested (e.g. 500 µg/mL) would be more informative.
Figure 2: The immunodetection of PAR is hardy visible in panel C. Are these basal levels? Why PAR levels after Bleomycin treatment are not shown?
Figure 3: Here PAR levels are shown after Bleo treatment, as detected with three different antibodies. However, the description is misleading since they indicate that significant results were obtained with anti-PAR from Enzo (lines 174-175, p. 5), while I can see that panel E (10H antibody; please quote correctly, not H10) shows a fluorescence intensity higher than that in panel N (with anti-PAR Enzo). Details of image processing with ImageJ software should be also provided, and in addition, an explanation of why one the most used PAR antibody (i.e. 10H) was not useful in VERO cells, should be given. Furthermore, the choice of cell fixation with formaldehyde should be justified, as other Authors have successfully used methanol-acetone as the best fixative for PAR detection (e.g., El-Khamisy et al., NAR, 31:5526, 2003).
Another important point in this Figure is that the efficacy of Olaparib shown in the PAR quantification in panel I, is rather poor. This raises questions about all other results. In Figure SB, the condition of Olaparib used is not clear, i.e. if PAR has been measured after Bleomycin or at basal levels.
Figure 4. The Comet test quantification (panel O) shows that Bleomycin has induced a very small amount of DNA damage, since the DDI is only 1.5 times higher than that of untreated cells. A radiomimetic drug should induce far more breaks. Also this point raises questions about the treatment conditions used in this work.
Minor points:
Figures with microscopy images lack of scale bar for magnification.
Reference list is not accurate with many errors (absent or incomplete titles, e.g. refs 11, 24, 39).
Some sentences in spanish were left in the Legends (e.g. SB).
Reviewer 3 Report
The manuscript “PARP Inhibitor Olaparib Causes No Potentiation of Bleomycin Effect in VERO cells, even in the Presence of Pooled ATM, DNA-PK and LigIV Inhibitors” by Valentina Perini et al. is presented as an article. In this article the authors show and discuss in an experimental and theoretical manner what makes the VERO cells resistant to Bleomycin and to a pooled of Inhibitors. While it contains some interesting information, several issues exist which should be corrected before any publication. I cannot recommend this manuscript for publication in the current form but hope the author will find my comments useful in rewriting the material for future publication.
Major points:
In general, the presentation of the results is very confusing. Some of the figures mentioned cannot be found such as: line 138 Fig. A2, 237 Fig. A4 etc.etc. Send the reader to unpublished material makes reading very very difficult, it remains mainly unsued. It may be necessary to identify and choose these figures and collected them in a new short document.
In Figure 3 the legend is not explained. Which are the differences between the Fig. 3B and C and between 3 E and F? The time or the bleomycin concentration?
Furthermore, the experiments conditions of the control (Fig.1, Fig.3 and Fig.4) are not specified. In my opinion is missing in each experiment, a control only with OLA (like blank), to verify the activity of OLA under the same experimental conditions and moreover the use of a mixture of bleomycin plus OLA at 25 nM in order to evaluate the effect of OLA in combination with bleomycin at different concentration.
Round 2
Reviewer 1 Report
The authors have addressed my comments appropiately. Just a small comment regarding the Sc-7150 antibody from Santa Cruz: As far as I know, it is an anti-PARP-1 rabbit antibody, however in the material and methods the authors indicate rabbit anti-PARP1/2.
Reviewer 2 Report
The revised manuscript by Perini and co-workers does not seem to have taken into account the criticisms raised previously, since they have not investigated any possible mechanism of resistance to Bleomycin making only speculations on possible reasons of this phenotype in VERO cells.
In particular, the evidence that PAR intensity after Bleomycin with Enzo antibody is very low (Fig. 3N, and that Olaparib does not abolish completely the fluorescence signal, as shown by quantification in Fig. 3 panel I (please define what y axis values represent), suggest that PARP activity is very low. In addition, measuring Olaparib activity only on very low PAR basal levels (Fig SC-panel B), makes this determination unreliable.
The Authors have now added the supplementary figure SD suggesting that indeed, PAR synthesis is low, at least judging from the Western blot showing PARylation after treatment with very high concentrations of H2O2, or MMS. The efficacy of Olaparib should be tested with Western blot after one of these treatments.
These low PAR levels indicate that either PARP protein levels are very low, or PARP activity is low. The PARP1/2 levels shown by fluorescence in Fig. 2 are not quantitative, since there is no comparison with other cell types known to express normal PARP1/2 levels. This comparison should be done by Western blot, which provides a quantitative evaluation of the PARP protein levels.
These levels, as well as the protein activity, could be another cause of lack of activity of Olaparib, that Authors have not considered in details, nor discussed. For this point see for instance, the recent reviews in DNA Repair (71:172,2018) and in Cancer Drug Resistance (2:608, 2019). As indicated in my previous review, the Reference list is inaccurate and not updated, and these more recent reviews should be cited.
The fixation with formaldehyde is important for cell morphology, that may be necessary for EM studies. For cytology, methanol provides a reasonable morphology, so that quantification of PAR chains would be more accurate. Studies quoted in the PhD thesis are not available, thus their citation is not useful.
Details for image processing have been provided only for image acquisition, but not for the use of Image J software: e.g., how the area of analysis was selected?
Legends to Figures are not revised (e.g. Fig SA).
Reviewer 3 Report
I believe that the Manuscript ID: ijms-747336 -has been significantly
improved and now warrants publication in IJMS
Round 3
Reviewer 2 Report
The unique modifications in the third version of the manuscript by Perini and co-workers is in the reference list, without any additional experiments.
The lines of evidence presented in their reply letter are not sufficient, since these refer to results obtained on T. cruzi PARP (not on VERO cells) and the parameters provided on VERO cells are only indirectly related to their PARP activity (proliferation stages of T. cruzi).
The problem with this manuscript are not the negative results. The problem is that the results presented are not sufficiently supported. In the absence of more quantitative and comparative analysis of PARP1/2 and PAR levels after Olaparib, by Western blot (a very simple experiment to perform), the results presented are not sufficient to warrant publication in Int. J. Mol. Sciences.